# SO(2)-EQUIVARIANT REINFORCEMENT LEARNING

**Dian Wang, Robin Walters, and Robert Platt**
Khoury College of Computer Sciences
Northeastern University
Boston, MA 02115, USA
{wang.dian, r.walters, r.platt}@northeastern.edu

## ABSTRACT

Equivariant neural networks enforce symmetry within the structure of their convolutional layers, resulting in a substantial improvement in sample efficiency when learning an equivariant or invariant function. Such models are applicable to robotic manipulation learning which can often be formulated as a rotationally symmetric problem. This paper studies equivariant model architectures in the context of $Q$-learning and actor-critic reinforcement learning. We identify equivariant and invariant characteristics of the optimal $Q$-function and the optimal policy and propose equivariant DQN and SAC algorithms that leverage this structure. We present experiments that demonstrate that our equivariant versions of DQN and SAC can be significantly more sample efficient than competing algorithms on an important class of robotic manipulation problems.

## 1 INTRODUCTION

A key challenge in reinforcement learning is to improve sample efficiency – that is to reduce the amount of environmental interactions that an agent must take in order to learn a good policy. This is particularly important in robotics applications where gaining experience potentially means interacting with a physical environment. One way of improving sample efficiency is to create "artificial" experiences through data augmentation. This is typically done in visual state spaces where an affine transformation (e.g., translation or rotation of the image) is applied to the states experienced during a transition (Laskin et al., 2020a; Kostrikov et al., 2020). These approaches implicitly assume that the transition and reward dynamics of the environment are invariant to affine transformations of the visual state. In fact, some approaches explicitly use a contrastive loss term to induce the agent to learn translation-invariant feature representations (Laskin et al., 2020b; Zhan et al., 2020).

Recent work in geometric deep learning suggests that it may be possible to learn transformation-invariant policies and value functions in a different way, using equivariant neural networks (Cohen & Welling, 2016a;b). The key idea is to structure the model architecture such that it is constrained only to represent functions with the desired invariance properties. In principle, this approach aim at exactly the same thing as the data augmentation approaches described above – both methods seek to improve sample efficiency by introducing an inductive bias. However, the equivariance approach achieves this more directly by modifying the model architecture rather than by modifying the training data. Since with data augmentation, the model must learn equivariance in addition to the task itself, more training time and greater model capacity are often required. Even then, data augmentation results only in approximate equivariance whereas equivariant neural networks guarantee it and often have stronger generalization as well (Wang et al., 2020b). While equivariant architectures have recently been applied to reinforcement learning (van der Pol et al., 2020a;b; Mondal et al., 2020), this has been done only in toy settings (grid worlds, etc.) where the model is equivariant over small finite groups, and the advantages of this approach over standard methods is less clear.

This paper explores the application of equivariant methods to more realistic problems in robotics such as object manipulation. We make several contributions. First, we define and analyze an important class of MDPs that we call *group-invariant MDPs*. Second, we introduce a new variation of the Equivariant DQN (Mondal et al., 2020), and we further introduce equivariant variations of SAC (Haarnoja et al., 2018), and learning from demonstration (LfD). Finally, we show that our

methods convincingly outperform recent competitive data augmentation approaches (Laskin et al., 2020a; Kostrikov et al., 2020; Laskin et al., 2020b; Zhan et al., 2020). Our Equivariant SAC method, in particular, outperforms these baselines so dramatically (Figure 7) that it could make reinforcement learning feasible for a much larger class of robotics problems than is currently the case. Supplementary video and code are available at `https://pointw.github.io/equi_rl_page/`.

## 2 RELATED WORK

**Equivariant Learning:** Encoding symmetries in the structure of neural networks can improve both generalization and sample efficiency. The idea of equivariant learning is first introduced in G-Convolution (Cohen & Welling, 2016a). The extension work proposes an alternative architecture, Steerable CNN (Cohen & Welling, 2016b). Weiler & Cesa (2019) proposes a general framework for implementing E(2)-Steerable CNNs. In the context of reinforcement learning, Mondal et al. (2020) investigates the use of Steerable CNNs in the context of two game environments. van der Pol et al. (2020b) proposes MDP homomorphic networks to encode rotational and reflectional equivariance of an MDP but only evaluates their method in a small set of tasks. In robotic manipulation, Wang et al. (2021) learns equivariant $Q$-functions but is limited in the spatial action space. In contrast to prior work, this paper proposes an Equivariant SAC algorithm, an equivariant LfD algorithm, and a novel variation of Equivariant DQN (Mondal et al., 2020) focusing on visual motor control problems.

**Data Augmentation:** Another popular method for improving sample efficiency is data augmentation. Recent works demonstrate that the use of simple data augmentation methods like random crop or random translate can significantly improve the performance of reinforcement learning (Laskin et al., 2020a; Kostrikov et al., 2020). Data augmentation is often used for generating additional samples (Kalashnikov et al., 2018; Lin et al., 2020; Zeng et al., 2020) in robotic manipulation. However, data augmentation methods are often less sample efficient than equivariant networks because the latter injects an inductive bias to the network architecture.

**Contrastive Learning:** Data augmentation is also applied with contrastive learning (Oord et al., 2018) to improve feature extraction. Laskin et al. (2020b) show significant sample-efficiency improvement by adding an auxiliary contrastive learning term using random crop augmentation. Zhan et al. (2020) use a similar method in the context of robotic manipulation. However, contrastive learning is limited to learning an invariant feature encoder and is not capable of learning equivariant functions.

**Close-Loop Robotic Control:** There are two typical action space definitions when learning policies that control the end-effector of a robot arm: the spatial action space that controls the target pose of the end-effector (Zeng et al., 2018b;a; Satish et al., 2019; Wang et al., 2020a), or the close-loop action space that controls the displacement of the end-effector. The close-loop action space is widely used for learning grasping policies (Kalashnikov et al., 2018; Quillen et al., 2018; Breyer et al., 2019; James et al., 2019). Recently, some works also learn more complex policies than grasping (Viereck et al., 2020; Kilinc et al., 2019; Cabi et al., 2020; Zhan et al., 2020). This work extends prior works in the close-loop action space by using equivariant learning to improve the sample efficiency.

## 3 BACKGROUND

**SO(2) and $C_n$:** We will reason about rotation in terms of the group SO(2) and its cyclic subgroup $C_n \leq \mathrm{SO}(2)$. SO(2) is the group of continuous planar rotations $\{\mathrm{Rot}_\theta : 0 \leq \theta < 2\pi\}$. $C_n$ is the discrete subgroup $C_n = \{\mathrm{Rot}_\theta : \theta \in \{\frac{2\pi i}{n}|0 \leq i < n\}\}$ of rotations by multiples $\frac{2\pi}{n}$.

**$C_n$ actions:** A group $G$ may be equipped with an *action* on a set $X$ by specifying a map $\cdot : G \times X \to X$ satisfying $g_1 \cdot (g_2 \cdot x) = (g_1 g_2) \cdot x$ and $1 \cdot x = x$ for all $g_1, g_2 \in G, x \in X$. Note that closure, $gx \in X$, and invertibility, $g^{-1}gx = x$, follow immediately from the definition. We are interested in *actions* of $C_n$ which formalize how vectors or feature maps transform under rotation. The group $C_n$ acts in three ways that concern us (for a more comprehensive background, see Bronstein et al. (2021)):

1. $\mathbb{R}$ through the *trivial representation* $\rho_0$. Let $g \in C_n$ and $x \in \mathbb{R}$. Then $\rho_0(g)x = x$. For example, the trivial representation describes how pixel color/depth values change when an image is rotated, i.e. they do not change (Figure 1 left).

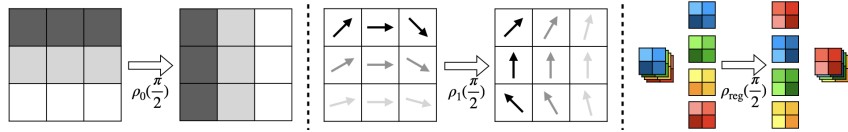

Figure 1: Illustration of how an element $g \in C_n$ acts on the feature map by rotating the pixels and transforming the channel space through $\rho_0$, $\rho_1$, and $\rho_{\text{reg}}$. Left: $C_n$ acts on the channel space of a 1-channel feature map by identical mapping. Middle: $C_n$ acts on the channel space of a vector field by rotating the vector at each pixel through $\rho_1$. Right: $C_n$ acts on the channel space of a 4-channel feature map by permuting the order of the channels by $\rho_{\text{reg}}$.

2. $\mathbb{R}^2$ through the *standard representation* $\rho_1$. Let $g \in C_n$ and $v \in \mathbb{R}^2$. Then $\rho_1(g)v = \left( \begin{smallmatrix} \cos g & -\sin g \\ \sin g & \cos g \end{smallmatrix} \right) v$. This describes how elements of a vector field change when rotated (Figure 1 middle).

3. $\mathbb{R}^n$ through the *regular representation* $\rho_{\text{reg}}$. Let $g = r^m \in C_n = \{1, r, r^2, \ldots, r^{n-1}\}$ and $(x_1, x_2, \ldots, x_n) \in \mathbb{R}^n$. Then $\rho_{reg}(g)x = (x_{n-m+1}, \ldots, x_n, x_1, x_2, \ldots, x_{n-m})$ cyclically permutes the coordinates of $\mathbb{R}^n$ (Figure 1 right).

Feature maps as functions: In deep learning, images and feature maps are typically expressed as tensors. However, it will be convenient here to sometimes express these as functions. Specifically, we may write an $h \times w$ one-channel image $\mathcal{F} \in \mathbb{R}^{1 \times h \times w}$ as a function $\mathcal{F} \colon \mathbb{R}^2 \to \mathbb{R}$ where $\mathcal{F}(x, y)$ describes the intensity at pixel $x, y$. Similarly, an $m$-channel tensor $\mathcal{F} \in \mathbb{R}^{m \times h \times w}$ may be written as $\mathcal{F} \colon \mathbb{R}^2 \to \mathbb{R}^m$. We refer to the domain of this function as its "spatial dimensions".

$C_n$ actions on vectors and feature maps: $C_n$ acts on vectors and feature maps differently depending upon their semantics. We formalize these different ways of acting as follows. Let $\mathcal{F} \colon \mathbb{R}^2 \to \mathbb{R}^m$ be an $m$-channel feature map and let $V \in \mathbb{R}^{m \times 1 \times 1} = \mathbb{R}^m$ be a vector represented as a special case of a feature map with $1 \times 1$ spatial dimensions. Then $g$ is defined to act on $\mathcal{F}$ by

$$(g\mathcal{F})(x, y) = \rho_j(g)\mathcal{F}(\rho_1(g)^{-1}(x, y)). \tag{1}$$

For a vector $V$ (considered to be at $(x, y) = (0, 0)$), this becomes:

$$gV = \rho_j(g)V. \tag{2}$$

In the above, $\rho_1(g)$ rotates pixel location and $\rho_j(g)$ transforms the pixel feature vector using the trivial representation ($\rho_j = \rho_0$), the standard representation ($\rho_j = \rho_1$), the regular representation ($\rho_j = \rho_{\text{reg}}$), or some combination thereof.

Equivariant convolutional layer: A $C_n$-equivariant layer is a function $h$ whose output is constrained to transform in a defined way when the input feature map is transformed by a group action. Consider an equivariant layer $h$ with an input $\mathcal{F}_{\text{in}} \colon \mathbb{R}^2 \to \mathbb{R}^{|\rho_{\text{in}}|}$ and an output $\mathcal{F}_{\text{out}} \colon \mathbb{R}^2 \to \mathbb{R}^{|\rho_{\text{out}}|}$, where $\rho_{\text{in}}$ and $\rho_{\text{out}}$ denote the group representations associated with $\mathcal{F}_{\text{in}}$ and $\mathcal{F}_{\text{out}}$, respectively. When the input is transformed, this layer is constrained to output a transformed version of the same output feature map:

$$h(g\mathcal{F}_{\text{in}}) = g(h(\mathcal{F}_{\text{in}})) = g\mathcal{F}_{\text{out}}. \tag{3}$$

where $g \in C_n$ acts on $\mathcal{F}_{\text{in}}$ or $\mathcal{F}_{\text{out}}$ through Equation 1 or Equation 2, i.e., this constraint equation can be applied to arbitrary feature maps $\mathcal{F}$ or vectors $V$.

A linear convolutional layer $h$ satisfies Equation 3 with respect to the group $C_n$ if the convolutional kernel $K \colon \mathbb{R}^2 \to \mathbb{R}^{|\rho_{\text{out}}| \times |\rho_{\text{in}}|}$ has the following form (Cohen et al., 2018):

$$K(\rho_1(g)v) = \rho_{\text{out}}^{-1}(g)K(v)\rho_{\text{in}}(g). \tag{4}$$

Since the composition of equivariant maps is equivariant, a fully convolutional equivariant network can be constructed by stacking equivariant convolutional layers that satisfy the constraint of Equation 3 and together with equivariant non-linearities (Weiler & Cesa, 2019).

## 4 PROBLEM STATEMENT

### 4.1 GROUP-INVARIANT MDPS

In a group-invariant MDP, the transition and reward functions are invariant to group elements $g \in G$ acting on the state and action space. For state $s \in S$, action $a \in A$, and $g \in G$, let $gs \in S$ denote the action of $g$ on $s$ and $ga \in A$ denote the action of $g$ on $a$.

**Definition 4.1** ($G$-invariant MDP). *A $G$-invariant MDP $\mathcal{M}_G = (S, A, T, R, G)$ is an MDP $\mathcal{M} = (S, A, T, R)$ that satisfies the following conditions:*

*1.* Reward Invariance: *The reward function is invariant to the action of the group element $g \in G$, $R(s, a) = R(gs, ga)$.*

*2.* Transition Invariance: *The transition function is invariant to the action of the group element $g \in G$, $T(s, a, s') = T(gs, ga, gs')$.*

A key feature of a $G$-invariant MDP is that its optimal solution is also $G$-invariant (proof in Appendix A):

**Proposition 4.1.** *Let $\mathcal{M}_G$ be a group-invariant MDP. Then its optimal Q-function is group invariant, $Q^*(s, a) = Q^*(gs, ga)$, and its optimal policy is group-equivariant, $\pi^*(gs) = g\pi^*(s)$, for any $g \in G$.*

It should be noted that the $G$-invariant MDP of Definition 4.1 is in fact a special case of an MDP homomorphism (Ravindran & Barto, 2001; 2004), a broad class of MDP abstractions. MDP homomorphisms are important because optimal solutions to the abstract problem can be "lifted" to produce optimal solutions to the original MDP (Ravindran & Barto, 2004). As such, Proposition 4.1 follows directly from those results.

### 4.2 SO(2)-INVARIANT MDPS IN VISUAL STATE SPACES

In the remainder of this paper, we focus exclusively on an important class of $\mathrm{SO}(2)$-invariant MDPs where the state is encoded as an image. We approximate $\mathrm{SO}(2)$ by its subgroup $C_n$.

State space: State is expressed as an $m$-channel image, $\mathcal{F}_s : \mathbb{R}^2 \to \mathbb{R}^m$. The group operator $g \in C_n$ acts on this image as defined in Equation 1 where we set $\rho_j = \rho_0$: $g\mathcal{F}_s(x, y) = \rho_0(g)\mathcal{F}_s(\rho_1(g)^{-1}(x, y))$, i.e., by rotating the pixels but leaving the pixel feature vector unchanged.

Action space: We assume we are given a factored action space $A_{\mathrm{inv}} \times A_{\mathrm{equiv}} = A \subseteq \mathbb{R}^k$ embedded in a $k$-dimensional Euclidean space where $A_{\mathrm{inv}} \subseteq \mathbb{R}^{k_{\mathrm{inv}}}$ and $A_{\mathrm{equiv}} \subseteq \mathbb{R}^{k - k_{\mathrm{inv}}}$. We require the variables in $A_{\mathrm{inv}}$ to be invariant with the rotation operator and the variables in $A_{\mathrm{equiv}}$ to rotate with the representation $\rho_{\mathrm{equiv}} = \rho_1$. Therefore, the rotation operator $g \in C_n$ acts on $a \in A$ via $ga = (\rho_{\mathrm{equiv}}(g)a_{\mathrm{equiv}}, a_{\mathrm{inv}})$ where $a_{\mathrm{inv}} \in A_{\mathrm{inv}}$ and $a_{\mathrm{equiv}} \in A_{\mathrm{equiv}}$.

Application to robotic manipulation: We express the state as a depth image centered on the gripper position where depth is defined relative to the gripper. The orientation of this image is relative to the base reference frame – not the gripper frame. We require the fingers of the gripper and objects grasped by the gripper to be visible in the image. Figure 2 shows an illustration. The action is a tuple, $a = (a_\lambda, a_{xy}, a_z, a_\theta) \in A \subset \mathbb{R}^5$, where $a_\lambda \in A_\lambda$ denotes the commanded gripper aperture, $a_{xy} \in A_{xy}$ denotes the commanded change in gripper $xy$ position, $a_z \in A_z$ denotes the commanded change in gripper height, and

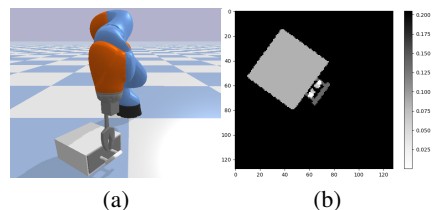

(a)         (b)

Figure 2: The manipulation scene (a) and the visual state space (b).

$a_\theta \in A_\theta$ denotes the commanded change in gripper orientation. Here, the $xy$ action is equivariant with $g \in C_n$, $A_{\mathrm{equiv}} = A_{xy}$, and the rest of the action variables are invariant, $A_{\mathrm{inv}} = A_\lambda \times A_z \times A_\theta$. Notice that the transition dynamics are $C_n$-invariant (i.e. $T(s, a, s') = T(gs, ga, gs')$) because the Newtonian physics of the interaction are invariant to the choice of reference frame. If we constrain the reward function to be $C_n$-invariant as well, then the resulting MDP is $C_n$-invariant.

## 5 APPROACH

### 5.1 EQUIVARIANT DQN

In DQN, we assume we have a discrete action space, and we learn the parameters of a $Q$-network that maps from the state onto action values. Given a $G$-invariant MDP, Proposition 4.1 tells us that the optimal $Q$-function is $G$-invariant. Therefore, we encode the $Q$-function using an equivariant neural network that is constrained to represent only $G$-invariant $Q$-functions. First, in order to use DQN, we need to discretize the action space. Let $\mathcal{A}_{\text{equiv}} \subset A_{\text{equiv}}$ and $\mathcal{A}_{\text{inv}} \subset A_{\text{inv}}$ be discrete subsets of the full equivariant and invariant action spaces, respectively. Next, we define a function $\mathcal{F}_a : \mathcal{A}_{\text{equiv}} \to \mathbb{R}^{\mathcal{A}_{\text{inv}}}$ from the equivariant action variables in $\mathcal{A}_{\text{equiv}}$ to the $Q$ values of the invariant action variables in $\mathcal{A}_{\text{inv}}$. For example, in the robotic manipulation domain described Section 4.2, we have $A_{\text{equiv}} = A_{xy}$ and $A_{\text{inv}} = A_\lambda \times A_z \times A_\theta$ and $\rho_{\text{equiv}} = \rho_1$, and we define $\mathcal{A}_{\text{equiv}}$ and $\mathcal{A}_{\text{inv}}$ accordingly. We now encode the $Q$ network $q$ as a stack of equivariant layers that each encode the equivariant constraint of Equation 3. Since the composition of equivariant layers is equivariant, $q$ satisfies:

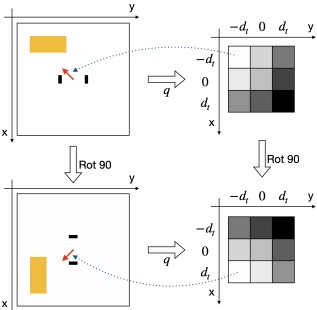

Figure 3: Illustration of $Q$-map equivariance. The output $Q$-map rotates with the input image.

$$q(g\mathcal{F}_s) = g(q(\mathcal{F}_s)) = g\mathcal{F}_a, \tag{5}$$

where we have substituted $\mathcal{F}_{\text{in}} = \mathcal{F}_s$ and $\mathcal{F}_{\text{out}} = \mathcal{F}_a$. In the above, the rotation operator $g \in C_n$ is applied using Equation 1 as $g\mathcal{F}_a(a_{xy}) = \rho_0(g)\mathcal{F}_a(\rho_1(g)^{-1}(a_{xy}))$. Figure 3 illustrates this equivariance constraint for the robotic manipulation example with $|\mathcal{A}_{\text{equiv}}| = |\mathcal{A}_{xy}| = 9$. When the state (represented as an image on the left) is rotated by 90 degrees, the values associated with the action variables in $\mathcal{A}_{xy}$ are also rotated similarly. The detailed network architecture is shown in Appendix D.1. Our architecture is different from that in Mondal et al. (2020) in that we associate the action of $g$ on $\mathcal{A}_{\text{equiv}}$ and $\mathcal{A}_{\text{inv}}$ with the group action on the spatial dimension and the channel dimension of a feature map $\mathcal{F}_a$, which is more efficient than learning such mapping using FC layers.

### 5.2 EQUIVARIANT SAC

In SAC, we assume the action space is continuous. We learn the parameters for two networks: a policy network $\Pi$ (the actor) and an action-value network $Q$ (the critic) (Haarnoja et al., 2018). The critic $Q : S \times A \to \mathbb{R}$ approximates $Q$ values in the typical way. However, the actor $\Pi : S \to A \times A_\sigma$ estimates both the mean and standard deviation of action for a given state. Here, we define $A_\sigma = \mathbb{R}^k$ to be the domain of the standard deviation variables over the $k$-dimensional action space defined in Section 4.2. Since Proposition 4.1 tells us that the optimal $Q$ is invariant and the optimal policy is equivariant, we must model $Q$ as an invariant network and $\Pi$ as an equivariant network.

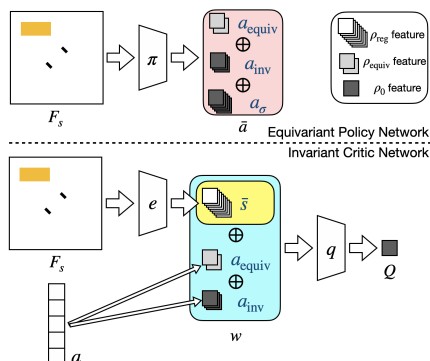

Figure 4: Illustration of the equivariant actor network (top) and the invariant critic network (bottom).

Policy network: First, consider the equivariant constraint of the policy network. As before, the state is encoded by the function $\mathcal{F}_s$. However, we must now express the action as a vector over $\bar{A} = A \times A_\sigma$. Factoring $A$ into its equivariant and invariant components, we have $\bar{A} = A_{\text{equiv}} \times A_{\text{inv}} \times A_\sigma$. In order to identify the equivariance relation for $\bar{A}$, we must define how the group operator $g \in G$ acts on $a_\sigma \in A_\sigma$. Here, we make the simplifying assumption that $a_\sigma$ is invariant to the group operator. This choice makes sense in robotics domains where we would expect the variance of our policy to be invariant to the choice of reference frame. As a result, we have that the group element $g \in G$ acts on $\bar{a} \in \bar{A}$ via:

$$g\bar{a} = g(a_{\text{equiv}}, a_{\text{inv}}, a_\sigma) = (\rho_{\text{equiv}}(g)a_{\text{equiv}}, a_{\text{inv}}, a_\sigma). \tag{6}$$

We can now define the actor network $\pi$ to be a mapping $\mathcal{F}_s \mapsto \bar{a}$ (Figure 4 top) that satisfies the following equivariance constraint (Equation 3):

$$\pi(g\mathcal{F}_s) = g(\pi(\mathcal{F}_s)) = g\bar{a}. \tag{7}$$

Critic network: The critic network takes both state and action as input and maps onto a real value. We define two equivariant networks: a state encoder $e$ and a $Q$ network $q$. The equivariant state encoder, $e$, maps the input state $\mathcal{F}_s$ onto a regular representation $\bar{s} \in (\mathbb{R}^n)^\alpha$ where each of $n$ group elements is associated with an $\alpha$-vector. Since $\bar{s}$ has a regular representation, we have $g\bar{s} = \rho_{\text{reg}}(g)\bar{s}$. Writing the equivariance constraint of Equation 3 for $e$, we have that $e$ must satisfy $e(g\mathcal{F}_s) = ge(\mathcal{F}_s) = g\bar{s}$. The output state representation $\bar{s}$ is concatenated with the action $a \in A$, producing $w = (\bar{s}, a)$. The action of the group operator is now $gw = (g\bar{s}, ga)$ where $ga = (\rho_{\text{equiv}}(g)a_{\text{equiv}}, a_{\text{inv}})$. Finally, the $q$ network maps from $w$ onto $\mathbb{R}$, a real-valued estimate of the $Q$ value for $w$. Based on proposition 4.1, this network must be invariant to the group action: $q(gw) = q(w)$. All together, the critic satisfies the following invariance equation:

$$q(e(g\mathcal{F}_s), ga) = q(e(\mathcal{F}_s), a). \tag{8}$$

This network is illustrated at the bottom of Figure 4. For a robotic manipulation domain in Section 4.2, we have $A_{\text{equiv}} = A_{xy}$ and $A_{\text{inv}} = A_\lambda \times A_z \times A_\theta$ and $\rho_{\text{equiv}} = \rho_1$. The detailed network architecture is in Appendix D.2.

Preventing the critic from becoming overconstrained: In the model architecture above, the hidden layer of $q$ is represented using a vector in the regular representation and the output of $q$ is encoded using the trivial representation. However, Schur's Lemma (see e.g. Dummit & Foote (1991)) implies there only exists a one-dimensional space of linear mappings from a regular representation to a trivial representation (i.e., $x = a\sum_i v_i$ where $x$ is a trivial representation, $a$ is a constant, and $v$ is a regular representation). This implies that a linear mapping $f : \mathbb{R}^n \times \mathbb{R}^n \to \mathbb{R}$ from two regular representations to a trivial representation that satisfies $f(gv, gw) = f(v, w)$ for all $g \in G$ will also satisfy $f(g_1 v, w) = f(v, w)$ and $f(v, g_2 w) = f(v, w)$ for all $g_1, g_2 \in G$. (See details in Appendix B.) In principle, this could overconstrain the last layer of $q$ to encode additional undesired symmetries. To avoid this problem we use a *non-linear* equivariant mapping, $\texttt{maxpool}$, over the group space to transform the regular representation to the trivial representation.

## 5.3 Equivariant SACfD

Many of the problems we want to address cannot be solved without guiding the agent's exploration somehow. In order to evaluate our algorithms in this context, we introduce the following simple strategy for learning from demonstration with SAC. First, prior to training, we pre-populate the replay buffer with a set of expert demonstrations generated using a hand-coded planner. Second, we introduce the following L2 term into the SAC actor's loss function:

$$\mathcal{L}_{\text{actor}} = \mathcal{L}_{\text{SAC}} + \mathbb{1}_e \left[ \frac{1}{2}((a \sim \pi(s)) - a_e)^2 \right], \tag{9}$$

where $\mathcal{L}_{\text{SAC}}$ is the actor's loss term in standard SAC, $\mathbb{1}_e = 1$ if the sampled transition is an expert demonstration and 0 otherwise, $a \sim \pi(s)$ is an action sampled from the output Gaussian distribution of $\pi(s)$, and $a_e$ is the expert action. Since both the sampled action $a \sim \pi(s)$ and the expert action $a_e$ transform equivalently, $\mathcal{L}_{\text{actor}}$ is compatible with the equivariance we introduce in Section 5.2. We refer to this method as SACfD (SAC from Demonstration).

## 6 Experiments

We evaluate Equivariant DQN and Equivariant SAC in the manipulation tasks shown in Figure 5. These tasks can be formulated as $SO(2)$-invariant MDPs. All environments have sparse rewards (+1 when reaching the goal and 0 otherwise). See environment details in Appendix C.

## 6.1 Equivariant DQN

We evaluate Equivariant DQN in the Block Pulling, Object Picking, and Drawer Opening tasks for the group $C_4$. The discrete action space is $A_\lambda = \{\text{OPEN, CLOSE}\}$; $A_{xy} = \{(x,y)|x, y \in$

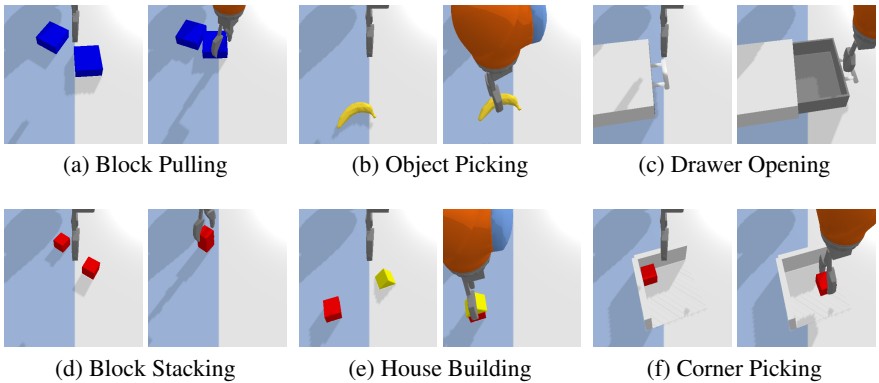

(a) Block Pulling        (b) Object Picking        (c) Drawer Opening

(d) Block Stacking        (e) House Building        (f) Corner Picking

Figure 5: The experimental environments implemented in PyBullet (Coumans & Bai, 2016). The left image in each sub figure shows an initial state of the environment; the right image shows the goal state. The poses of the objects are randomly initialized.

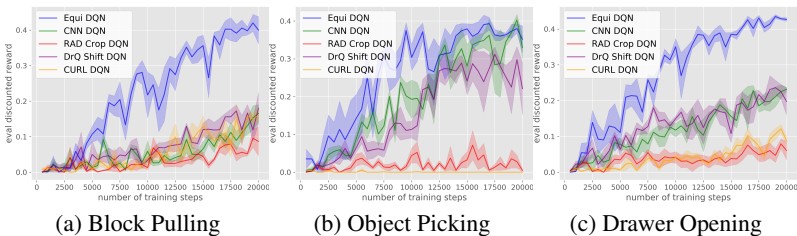

(a) Block Pulling        (b) Object Picking        (c) Drawer Opening

Figure 6: Comparison of Equivariant DQN (blue) with baselines. The plots show the evaluation performance of the greedy policy in terms of the discounted reward. The evaluation is performed every 500 training steps. Results are averaged over four runs. Shading denotes standard error.

$\{-0.02m, 0m, 0.02m\}\}$; $A_z = \{-0.02m, 0m, 0.02m\}$; $A_\theta = \{-\frac{\pi}{16}, 0, \frac{\pi}{16}\}$. Note that the definition of $A_{xy}$ and $g \in C_4$ satisfies the closure requirement of the action space in a way that $\forall a_{xy} \in A_{xy}, \forall g \in C_4, \rho_1(g)a_{xy} \in A_{xy}$. We compare Equivariant DQN (Equi DQN) against the following baselines: 1) CNN DQN: DQN with conventional CNN instead of equivariant network, where the conventional CNN has a similar amount of trainable parameters (3.9M) as the equivariant network (2.6M). 2) RAD Crop DQN (Laskin et al., 2020a): same network architecture as CNN DQN. At each training step, each transition in the minibatch is applied with a random-crop data augmentation. 3) DrQ Shift DQN (Kostrikov et al., 2020): same network architecture as CNN DQN. At each training step, both the $Q$-targets and the TD losses are calculated by averaging over two random-shift augmented transitions. 4): CURL DQN (Laskin et al., 2020b): similar architecture as CNN DQN with an extra contrastive loss term that learns an invariant encoder from random crop augmentations. See the baselines detail in Appendix E. At the beginning of each training process, we pre-populate the replay buffer with 100 episodes of expert demonstrations.

Figure 6 compares the learning curves of the various methods. Equivariant DQN learns faster and converges at a higher discounted reward in all three environments.

## 6.2 EQUIVARIANT SAC

In this experiment, we evaluate the performance of Equivariant SAC (Equi SAC) for the group $C_8$. The continuous action space is: $A_\lambda = [0, 1]$; $A_{xy} = \{(x, y)|x, y \in [-0.05m, 0.05m]\}$; $A_z = [-0.05m, 0.05m]$; $A_\theta = [-\frac{\pi}{8}, \frac{\pi}{8}]$. We compare against the following baselines: 1) CNN SAC: SAC with conventional CNN rather than equivariant networks, where the conventional CNN has a similar amount of trainable parameters (2.6M) as the equivariant network (2.3M). 2) RAD Crop SAC (Laskin et al., 2020a): same model architecture as CNN SAC with random crop data augmentation when sampling transitions. 3) DrQ Shift SAC (Kostrikov et al., 2020): same model

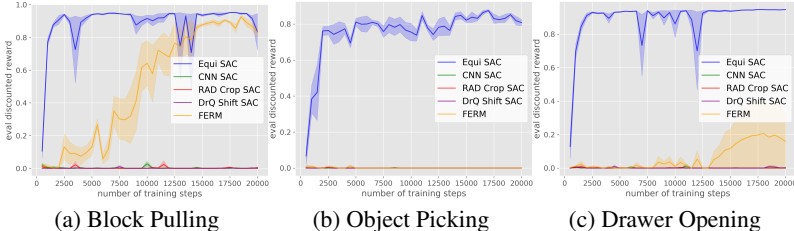

(a) Block Pulling      (b) Object Picking      (c) Drawer Opening

Figure 7: Comparison of Equivariant SAC (blue) with baselines. The plots show the evaluation performance of the greedy policy in terms of the discounted reward. The evaluation is performed every 500 training steps. Results are averaged over four runs. Shading denotes standard error.

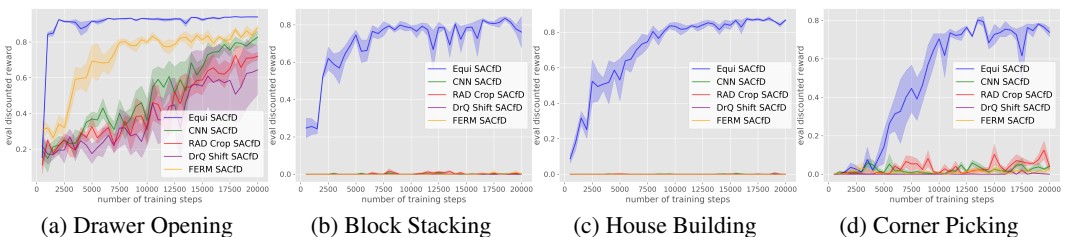

(a) Drawer Opening    (b) Block Stacking    (c) House Building    (d) Corner Picking

Figure 8: Comparison of Equivariant SACfD (blue) with baselines. The plots show the evaluation performance of the greedy policy in terms of the discounted reward. The evaluation is performed every 500 training steps. Results are averaged over four runs. Shading denotes standard error.

architecture as CNN SAC with random shift data augmentation when calculating the $Q$-target and the loss. 4) FERM (Zhan et al., 2020): a combination of SAC, contrastive learning, and random crop augmentation (baseline details in Appendix E). All methods use a $SO(2)$ data augmentation buffer, where every time a new transition is added, we generate 4 more augmented transitions by applying random continuous rotations to both the image and the action (this data augmentation in the buffer is in addition to the data augmentation that is performed in the RAD DrQ, and FERM baselines). Prior to each training run, we pre-load the replay buffer with 20 episodes of expert demonstration.

Figure 7 shows the comparison among the various methods. Notice that Equivariant SAC outperforms the other methods significantly. Without the equivariant approach, Object Picking and Drawer Opening appear to be infeasible for the baseline methods. In Block Pulling, FERM is the only other method able to solve the task.

## 6.3 EQUIVARIANT SACFD

We want to explore our equivariant methods in the context of more challenging tasks such as those in the bottom row of Figure 5. However, since these tasks are too difficult to solve without some kind of guided exploration, we augment the Equivariant SAC as well as all the baselines in two ways: 1) we use SACfD as described in Section 5.3; 2) we use Prioritized Experience Replay (Schaul et al., 2015) rather than standard replay buffer. As in Section 6.2, we use the $SO(2)$ data augmentation in the buffer that generates 4 extra $SO(2)$-augmented transitions whenever a new transition is added. Figure 8 shows the results. First, note that our Equivariant SACfD does best on all four tasks, followed by FERM, and other baselines. Second, notice that only the equivariant method can solve the last three (most challenging tasks). This suggests that equivariant models are important not only for unstructured reinforcement learning, but also for learning from demonstration. Additional results for Block Pulling and Object Picking environments are shown in Appendix G.

## 6.4 COMPARING WITH LEARNING EQUIVARIANCE USING AUGMENTATION

In the previous experiments, we compare against the data augmentation baselines using the same data augmentation operators that the authors proposed (random crop in RAD (Laskin et al., 2020a)

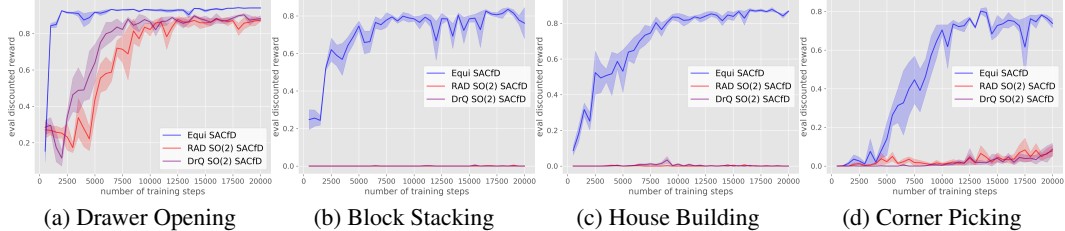

|  (a) Drawer Opening | (b) Block Stacking | (c) House Building | (d) Corner Picking |

Figure 9: Comparison of Equivariant SACfD (blue) with baselines. The plots show the evaluation performance of the greedy policy in terms of the discounted reward. The evaluation is performed every 500 training steps. Results are averaged over four runs. Shading denotes standard error.

and random shift in DrQ (Kostrikov et al., 2020)). However, those two methods can also be modified to learn $SO(2)$ equivariance using $SO(2)$ data augmentation. Here, we explore this idea as an alternative to our equivariant model. Specifically, instead of augmenting on the state as in Laskin et al. (2020a) and Kostrikov et al. (2020) using only translation, we apply the $SO(2)$ augmentation in both the state and the action. Since the RAD and DrQ baselines in this section are already running $SO(2)$ augmentations themselves, we disable the $SO(2)$ buffer augmentation for the online transitions in those baselines. (See the result of RAD and DrQ with the $SO(2)$ data augmentation buffer in Appendix H.4). We compare the resulting version of RAD (RAD $SO(2)$ SACfD) and DrQ (DrQ $SO(2)$ SACfD) with our Equivariant SACfD in Figure 9. Our method outperforms both RAD and DrQ equipped with $SO(2)$ data augmentation. Additional results for Block Pulling and Object Picking are shown in Appendix G.

## 6.5 GENERALIZATION EXPERIMENT

This experiment evaluates the ability for the equivariant model to generalize over the equivariance group. We use a similar experimental setting as in Section 6.3. However, now the training environment is always initialized with a fixed orientation rather than a random orientation. For example, in Block Pulling, the two blocks are initialized with a fixed relative orientation; in Drawer Opening, the drawer is initialized with a fixed orientation. In the evaluation environment, however, these objects are initialized with random orientations. To suc-

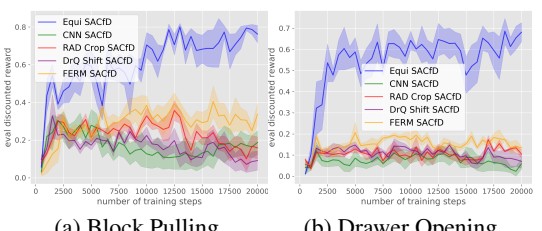

|  (a) Block Pulling | (b) Drawer Opening |

Figure 10: Comparison of Equivariant SACfD with baselines. Results are averaged over four runs.

ceed, the agent needs to generalize over varied orientations while being trained with a fixed orientation. To prevent the agent from generalizing via augmentation, we disable the $SO(2)$ augmentation in the buffer. As shown in Figure 10, Equivariant SACfD generalizes better than the baselines. Even though the equivariant network is presented with only one orientation during training, it successfully generalizes over random orientation whereas none of the baselines can.

## 7 DISCUSSION

This paper defines a class of group-invariant MDPs and identifies the invariance and equivariance characteristics of their optimal solutions. This paper further proposes Equivariant SAC and a new variation of Equivariant DQN for continuous action space and discrete action space, respectively. We show experimentally in the robotic manipulation domains that our proposal substantially surpasses the performance of competitive baselines. A key limitation of this work is that our definition of $G$-invariant MDPs requires the MDP to have an invariant reward function and invariant transition function. Though such restrictions are often applicable in robotics, they limit the potential of the proposed methods in other domains like some ATARI games. Furthermore, if the observation is from a non-top-down perspective, or there are non-equivariant structures in the observation (e.g., the robot arm), the invariant assumptions of a $G$-invariant MDP will not be directly satisfied.

ACKNOWLEDGMENTS

This work is supported in part by NSF 1724257, NSF 1724191, NSF 1763878, NSF 1750649, and NASA 80NSSC19K1474. R. Walters is supported by the Roux Institute and the Harold Alfond Foundation and NSF grants 2107256 and 2134178.

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

## A    PROOF OF PROPOSITION 4.1

The proof in this section follows Wang et al. (2021). Note that the definition of group action $\cdot : G \times X \to X$ implies that elements $g \in G$ act by bijections on $X$ since the action of $g^{-1}$ gives a two-sided inverse for the action of $g$. That is, $g$ permutes the elements of $X$.

*Proof of Proposition 4.1.* For $g \in G$, we will first show that the optimal $Q$-function is $G$-invariant, i.e., $Q^*(s, a) = Q^*(gs, ga)$, then show that the optimal policy is $G$-equivariant, i.e., $\pi^*(gs) = g\pi^*(s)$.

(1) $Q^*(s, a) = Q^*(gs, ga)$: The Bellman optimality equations for $Q^*(s, a)$ and $Q^*(gs, ga)$ are, respectively:

$$Q^*(s, a) = R(s, a) + \gamma \sup_{a' \in A} \int_{s' \in S} T(s, a, s') Q^*(s', a'), \tag{10}$$

and

$$Q^*(gs, ga) = R(gs, ga) + \gamma \sup_{a' \in A} \int_{s' \in S} T(gs, ga, s') Q^*(s', a'). \tag{11}$$

Since $g \in G$ merely permutes the elements of $S$, we can re-index the integral using $\bar{s}' = gs'$:

$$Q^*(gs, ga) = R(gs, ga) + \gamma \sup_{\bar{a}' \in gA} \int_{\bar{s}' \in gS} T(gs, ga, \bar{s}') Q^*(\bar{s}', \bar{a}') \tag{12}$$

$$= R(gs, ga) + \gamma \sup_{a' \in A} \int_{s' \in S} T(gs, ga, gs') Q^*(gs', ga'). \tag{13}$$

Using the Reward Invariance and the Transition Invariance in Definition 4.1, this can be written:

$$Q^*(gs, ga) = R(s, a) + \gamma \sup_{a' \in A} \int_{s' \in S} T(s, a, s') Q^*(gs', ga'). \tag{14}$$

Now, define a new function $\bar{Q}$ such that $\forall s, a \in S \times A$, $\bar{Q}(s, a) = Q(gs, ga)$ and substitute into Eq. 14, resulting in:

$$\bar{Q}^*(s, a) = R(s, a) + \gamma \sup_{a' \in A} \int_{s' \in S} T(s, a, s') \bar{Q}^*(s', a'). \tag{15}$$

Notice that Eq. 15 and Eq. 10 are the same Bellman equation. Since solutions to the Bellman equation are unique, we have that $\forall s, a \in S \times A$, $Q^*(s, a) = \bar{Q}^*(s, a) = Q^*(gs, ga)$.

(2) $\pi^*(gs) = g\pi^*(s)$: The optimal policy for $\pi^*(s)$ and $\pi^*(gs)$ can be written in terms of the optimal $Q$-function, $Q^*$, as:

$$\pi^*(s) = \arg\max_{a \in A} Q^*(s, a) \tag{16}$$

and

$$\pi^*(gs) = \arg\max_{\bar{a} \in A} Q^*(gs, \bar{a}) \tag{17}$$

Using the invariant property of $Q^*$ we can substitute $Q^*(gs, \bar{a})$ with $Q^*(s, g^{-1}\bar{a})$ in Equation 17:

$$\pi^*(gs) = \arg\max_{\bar{a} \in A} Q^*(s, g^{-1}\bar{a}) \tag{18}$$

Let $\bar{a} = ga$, Equation 18 can be written as:

$$\pi^*(gs) = g[\arg\max_{a \in A} Q^*(s, g^{-1}ga)] \tag{19}$$

Cancelling $g^{-1}$ and $g$ and substituting Equation 16 we have,

$$\pi^*(gs) = g\pi^*(s). \tag{20}$$

$\square$

## B    Equivariance Overconstrain

**Proposition B.1.** *Let $f : V_{reg} \oplus V_{reg} \to V_{triv}$ be a linear $C_n$-equivariant function. Then $f(v, w) = a \sum_i v_i + b \sum_i w_i$.*

*Proof.* By Weyl decomposibility (Hall, 2003), $V_{reg}$ decomposes into irreducible representations for $C_n$ each with multiplicity determined by its dimension. Among these is the trivial representation with multiplicity 1. By Schur's lemma (Dummit & Foote, 1991), the mapping $V_{reg} \oplus V_{reg} \to V_{triv}$ must factor through the trivial representation embedded in $V_{reg}$. The projection onto the trivial representation is given $v \mapsto a \sum_i v_i$. The result follows by linearity. $\square$

As a corollary, we find that $C_n$-equivariant maps $V_{reg} \oplus V_{reg} \to V_{triv}$ are actually $C_n \times C_n$-equivariant. Let $(g_1, g_2) \in C_n \times C_n$, then applying the Proposition $f(g_1 v, g_2 w) = a \sum_i (gv)_i + b \sum_i (gw)_i = a \sum_i v_i + b \sum_i w_i = f(v, w)$.

## C    Environment Details

In all environments, the environment reset is conduced by randomly initializing the objects with random positions and orientations inside the workspace. The arm is always initialized at the same configuration. The workspace has a size of $0.4m \times 0.4m \times 0.24m$. All environments have a sparse reward, i.e., the agent acquires a +1 reward when reaching the goal state, and 0 otherwise. In the Py-Bullet simulator, the robot joints have enough compliance to allow the gripper to apply force on the block in the Corner Picking task.

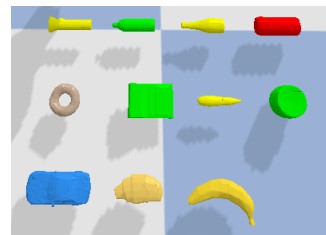

We augment the state image with an additional binary channel (i.e., either all pixels are 1 or all pixels are 0) indicating if the gripper is holding an object. Note that this additional channel is invariant to rotations (because all pixels have the same value) so it won't break the proposed equivariant properties.

Figure 11: The object set for Object Picking environment

The Block Pulling requires the robot to pull one block to make contact with the other block. The Object Picking requires the robot the pick up an object randomly sampled from a set of 11 objects (Figure 11). The Drawer Opening requires the robot to pull open a drawer. The Block Stacking requires the robot to stack one block on top of another. The House Building requires the robot to stack a triangle roof on top of a block. The Corner Picking requires the robot to slide the block from the corner and then pick it up.

## D    Network Architecture

Our equivariant models are implemented using the E2CNN (Weiler & Cesa, 2019) library with PyTorch (Paszke et al., 2017).

### D.1    Equivariant DQN Architecture

In the Equivariant DQN, we use a 7-layer Steerable CNN defined in the group $C_4$ (Figure 12a). The input $\mathcal{F}_s$ is encoded as a 2-channel $\rho_0$ feature map, and the output is a 18-channel $3 \times 3$ $\rho_0$ feature map where the channel encodes the invariant actions $\mathcal{A}_{\text{inv}}$ and the spatial dimension encodes $\mathcal{A}_{xy}$.

### D.2    Equivariant SAC Architecture

In the Equivariant SAC, there are two separate networks, both are Steerable CNN defined in the group $C_8$. The actor $\pi$ (Figure 12b top) is an 8-layer network that takes in a 2-channel $\rho_0$ feature map ($\mathcal{F}_s$) and outputs a mixed representation type $1 \times 1$ feature map ($\bar{a}$) consisting of 1 $\rho_1$ feature for $a_{xy}$ and 8 $\rho_0$ features for $a_{\text{inv}}$ and $a_\sigma$. The critic (Figure 12b bottom) is a 9-layer network that takes in both $\mathcal{F}_s$ as a 2-channel $\rho_0$ feature map and $a$ as a $1 \times 1$ mixed representation feature map

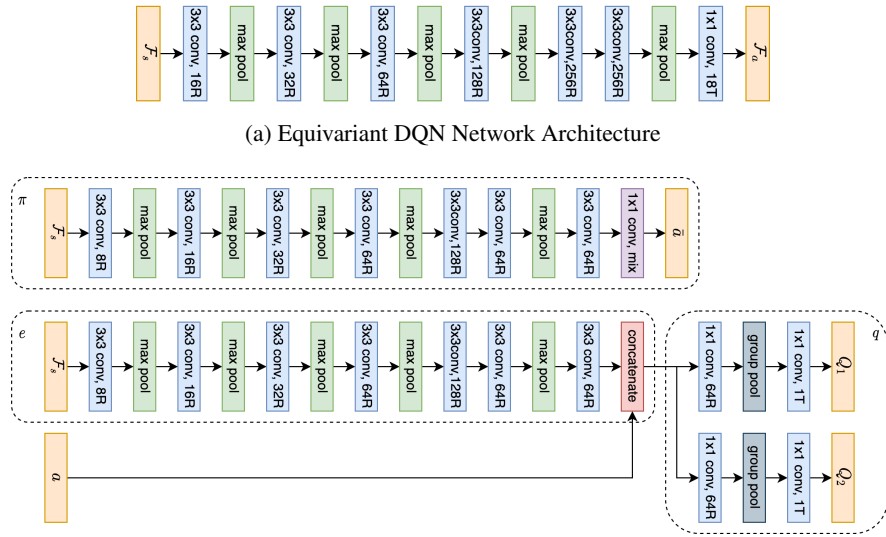

(a) Equivariant DQN Network Architecture

(b) Equivariant SAC Network Architecture

Figure 12: The architecture of the Equivariant DQN (a) and the Equivariant SAC (b). ReLU nonlinearity is omitted in the figure. A convolutional layer with a suffix of R indicates a regular representation layer (e.g., 16R is a 16-channel regular representation layer); a convolution layer with a suffix of T indicates a trivial representation layer (e.g., 1T is a 1-channel trivial representation layer).

consisting of 1 $\rho_1$ feature for $a_{xy}$ and 3 $\rho_0$ for $a_{\mathrm{inv}}$. The upper path $e$ encodes $\mathcal{F}_s$ into a 64-channel regular representation feature map $\bar{s}$ with $1 \times 1$ spatial dimensions, then concatenates it with $a$. Two separate $Q$-value paths $q$ take in the concatenated feature map and generate two $Q$-estimates in the form of $1 \times 1$ $\rho_0$ feature. The non-linear maxpool layer is used for transforming regular representations into trivial representations to prevent the equivariant overconstraint (Section 5.2). Note that there are two $Q$ outputs based on the requirement of the SAC algorithm.

## E    BASELINE DETAILS

Figure 13 shows the baseline network architectures for DQN and SAC. The RAD (Laskin et al., 2020a) Crop baselines, CURL (Laskin et al., 2020b) baselines, and FERM (Zhan et al., 2020) baselines use random crop for data augmentation. The random crop crops a $142 \times 142$ state image to the size of $128 \times 128$. The contrastive encoder of CURL baselines has a size of 128 as in Laskin et al. (2020b), and that for the FERM baselines has a size of 50 as in Zhan et al. (2020). The FERM baseline's contrastive encoder is pretrained for 1.6k steps using the expert data as in Zhan et al. (2020). The DrQ (Kostrikov et al., 2020) Shift baselines use random shift of $\pm 4$ pixels for data augmentation as in the original work. In all DrQ baselines, the number of augmentations for calculating the target $K$ and the number of augmentations for calculating the loss $M$ are both 2 as in Kostrikov et al. (2020).

## F    TRAINING DETAILS

We implement our experimental environments in the PyBullet simulator (Coumans & Bai, 2016). The workspace's size is $0.4m \times 0.4m \times 0.24m$. The pixel size of the visual state $I$ is $128 \times 128$ (except for the RAD Crop baselines, CURL baselines, and FERM baselines, where $I$'s size is $142 \times 142$ and will be cropped to $128 \times 128$). $I$'s FOV is $0.6m \times 0.6m$. During training, we use 5 parallel environments. We implement all training in PyTorch (Paszke et al., 2017). Both DQN and SAC use soft target update with $\tau = 10^{-2}$.

In the DQN experiments, we use the Adam optimizer (Kingma & Ba, 2014) with a learning rate of $10^{-4}$. We use Huber loss (Huber, 1964) for calculating the TD loss. We use a discount factor $\gamma = 0.95$. The batch size is 32. The buffer has a capacity of 100,000 transitions.

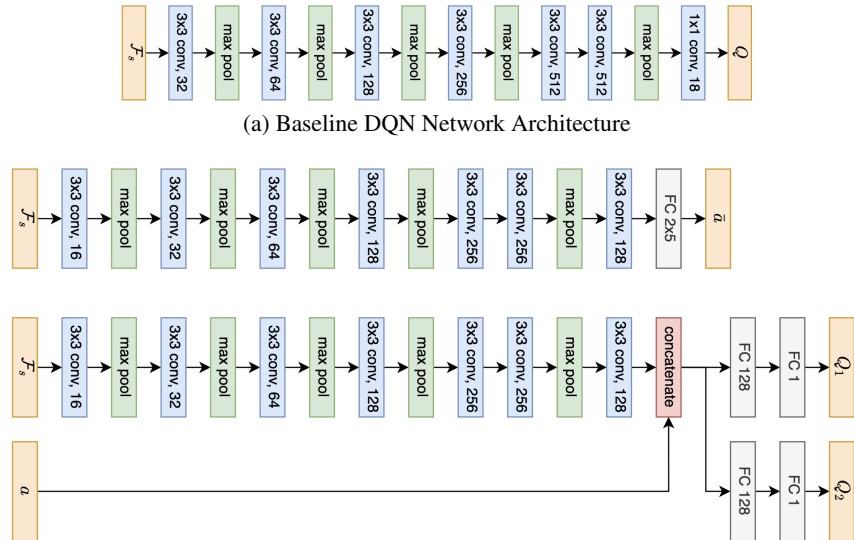

(a) Baseline DQN Network Architecture

(b) Baseline SAC Network Architecture

Figure 13: The architecture of the baseline conventional CNN DQN (a) and the baseline conventional CNN SAC (b). The baseline CNN architectures have similar amount of trainable parameters as the equivariant architectures. Specifically, Equivariant DQN has 2.6M parameters, and baseline DQN has 3.9M parameters; Equivariant SAC has 2.3M parameters, and baseline SAC has 2.6M parameters. ReLU nonlinearity is omitted in the figure.

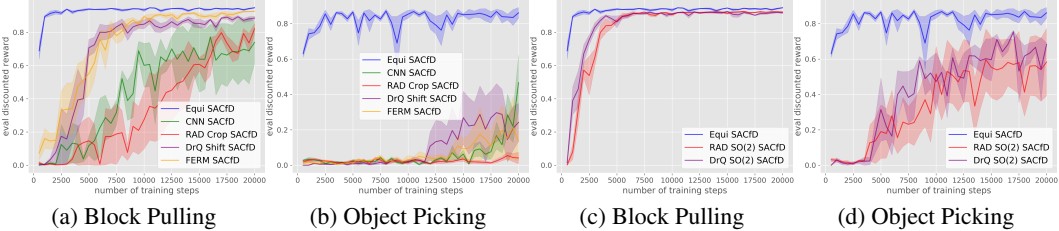

(a) Block Pulling      (b) Object Picking      (c) Block Pulling      (d) Object Picking

Figure 14: (a)-(b): additional results for Section 6.3. (c)-(d): additional results for Section 6.4. The plots show the evaluation performance of the greedy policy in terms of the discounted reward. The evaluation is performed every 500 training steps. Results are averaged over four runs. Shading denotes standard error.

In the SAC (and SACfD) experiments, we use the Adam optimizer with a learning rate of $10^{-3}$. The entropy temperature $\alpha$ is initialized at $10^{-2}$. The target entropy is -5. The discount factor $\gamma = 0.99$. The batch size is 64. The buffer has a capacity of 100,000 transitions. SACfD uses the prioritized replay buffer (Schaul et al., 2015) with prioritized replay exponent of 0.6 and prioritized importance sampling exponent $\beta_0 = 0.4$ as in Schaul et al. (2015). The expert transitions are given a priority bonus of $\epsilon_d = 1$.

## G   ADDITIONAL EXPERIMENTAL RESULTS FOR EQUIVARIANT SACFD

Figure 14 (a)-(b) shows the results for the experiment of Section 6.3 in Block Pulling and Object Picking environments. The Equivariant SACfD outperforms all baselines in those two environments.

Figure 14 (c)-(d) shows the results for the experiment of Section 6.4 in block Pulling and Object Picking environments. Similarly as the results in Figure 9, our Equivariant SACfD outperforms both RAD and DrQ equipped with SO(2) dat augmentation.

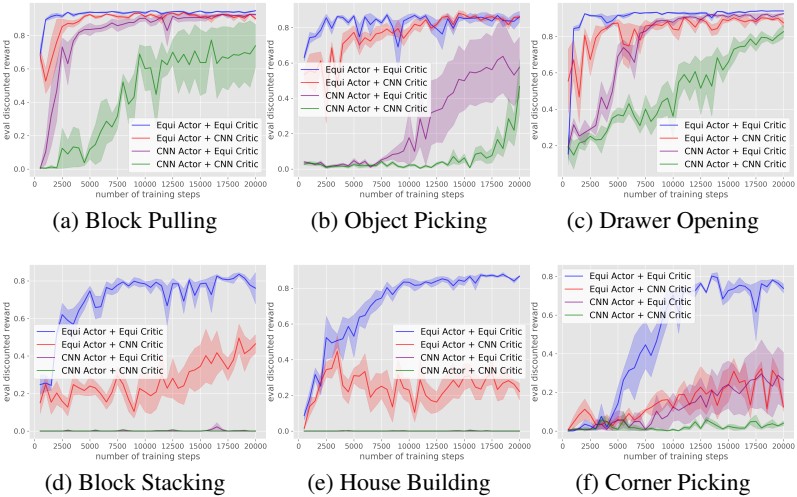

Figure 15: Ablation of using equivariant network solely in actor network or critic network. The plots show the evaluation performance in terms of discounted reward during training. The evaluation is performed every 500 training steps. Results are averaged over four runs. Shading denotes standard error.

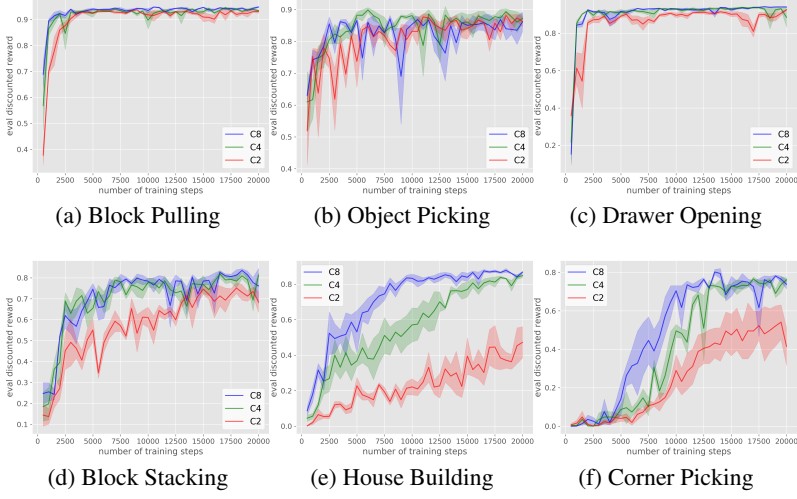

Figure 16: Ablation of using different symmetry groups ($C_8$, $C_4$, or $C_2$), in Equivariant SACfD. The plots show the evaluation performance of the greedy policy in terms of the discounted reward. The evaluation is performed every 500 training steps. Results are averaged over four runs. Shading denotes standard error.

## H    ABLATION STUDIES

### H.1    USING EQUIVARIANT NETWORK ONLY IN ACTOR OR CRITIC

In this experiment, we investigate the effectiveness of the equivariant network in SACfD by only applying it in the actor network or the critic network. We evaluate four variations: 1) Equi Actor + Equi Critic that uses equivariant network in both the actor and the critic; 2) Equi Actor + CNN Critic that uses equivariant network solely in the actor and uses conventional CNN in the critic; 3) CNN Actor + Equi Critic that uses conventional CNN in the actor and equivariant network in the Critic; 4) CNN Actor + CNN Critic that uses the conventional CNN in both the actor and the critic. Other experimental setup mirrors Section 6.3. As is shown in Figure 15, applying the equivariant network

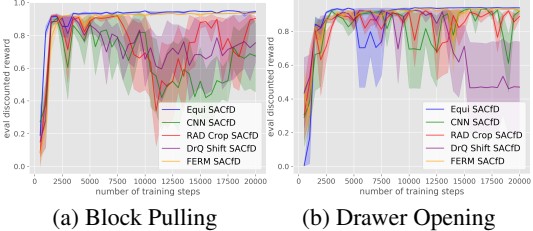

(a) Block Pulling (b) Drawer Opening

Figure 17: Ablation of using equivariant architecture in non-symmetric tasks. The plots show the evaluation performance of the greedy policy in terms of the discounted reward. The evaluation is performed every 500 training steps. Results are averaged over four runs. Shading denotes standard error.

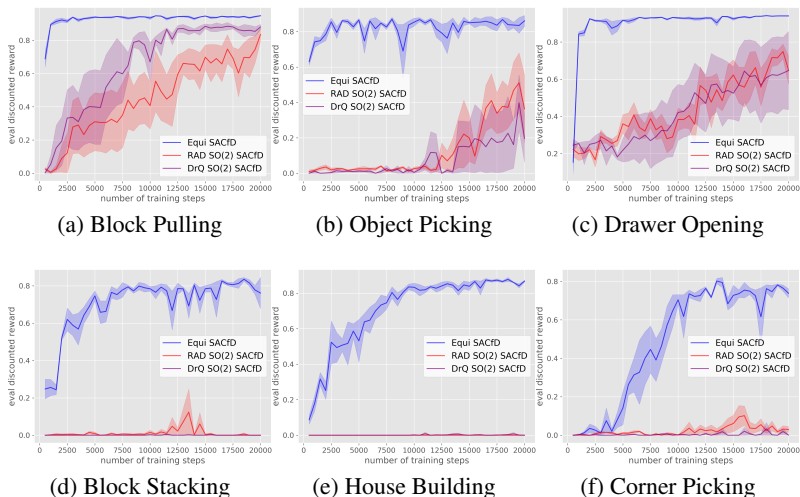

(a) Block Pulling (b) Object Picking (c) Drawer Opening

(d) Block Stacking (e) House Building (f) Corner Picking

Figure 18: Ablation of comparing against rotational augmentation baselines applied with rotational buffer augmentation. The plots show the evaluation performance of the greedy policy in terms of the discounted reward. The evaluation is performed every 500 training steps. Results are averaged over four runs. Shading denotes standard error.

in the actor generally helps more than applying the equivariant network in the critic (in 5 out of 6 experiments), and using the equivariant network in both the actor and the critic always demonstrates the best performance.

## H.2 DIFFERENT SYMMETRY GROUPS

This experiment compares the equivariant networks defined in three different symmetry groups: $C_8$, $C_4$, and $C_2$. We run this experiment in SACfD with the same setup as in Section 6.3. As is shown in Figure 16, the network defined in $C_8$ generally outperforms the network defined in $C_4$, followed by the network defined in $C_2$.

## H.3 EQUIVARIANT SACFD IN NON-SYMMETRIC ENVIRONMENTS

This experiments evaluates the performance of Equivariant SACfD in non-symmetric tasks where the initial orientation of the environments are fixed rather than random. (Similarly as in Section 6.5 but both the training and the evaluation environments have the fix orientation.) Specifically, in Block Pulling, the two blocks in the training environment is initialized with a fixed relative orientation; in Drawer Opening, the drawer is initialized with a fixed orientation. As is shown in Figure 17, when the environments do not contain $SO(2)$ symmetries, the performance gain of using equivariant network is less significant.

### H.4 ROTATIONAL AUGMENTATION + BUFFER AUGMENTATION

Section 6.4 compares our Equivariant SACfD with rotational data augmentation baselines. This experiment shows the performance of those baselines (and an extra CNN SACfD baseline that uses conventional CNN) equipped with the data augmentation buffer. As is mentioned in Section 6.2, the data augmentation baseline creates 4 extra augmented transitions using random $SO(2)$ rotation every time a new transition is added. Figure 18 shows the result, where none of the baselines outperform our proposal in any tasks. Compared with Figure 9, the data augmentation buffer hurts RAD and DrQ because of the redundancy of the same data augmentation.

