# OpenReview forum: "$\mathrm{SO}(2)$-Equivariant Reinforcement Learning"
_ICLR.cc/2022/Conference — ICLR 2022 Spotlight_

### Official Review · Reviewer_PtaK · 2021-11-02

**Correctness:** 3
**Technical Novelty And Significance:** 2
**Empirical Novelty And Significance:** 2
**Recommendation:** 5
**Confidence:** 4

**Main Review:**

Although I liked the paper, I would like to kindly ask the authors the following questions:

1. The authors have provided a definition of G-invariant MDPs and demonstrated their effectiveness in visual tasks. This definition rather introduces restrictions on the class of MDPs considered. My question is apart from visual tasks like those solved in this paper, can the authors provide examples and intuitions into when those assumptions hold? Is it the case that G-invariant MDPs are exclusive to image state representations?

2. Similar question as the above goes for the action spaces. What assumptions are we really making on the reward and transition functions? Do these relate to Lipshitzness at all? Can we handle collision-avoidance scenarios for example?

3. In the experiments, the authors have considered relatively low-dimensional action spaces. Can we please see a demonstration in higher dimensional action spaces? Would the method apply to ATARI domains as well? Or is it that the type of discontinuities present in ATARI prohibit the application of the proposed method? It would be great to have a section discussing the limitations in terms of MPDs that could be considered when using the current approach.

4. Although a minor point, I am a bit confused about the novelty of the current approach beyond applying equivariant architectures to reinforcement learning. Can the authors please help me understand the major contributions of this work?

**Summary Of The Paper:**

In this paper, the authors present an application of equivariant neural networks to reinforcement learning. They demonstrate new DQN and SAC architectures that make use of equivariant representations in order to improve sample efficiency. Results on manipulation tasks demonstrate sample complexity reduction compared to other techniques.

**Summary Of The Review:**

Please see above.

---

> ### Author Response · Authors · 2021-11-17
> **Author Response to Reviewer PtaK**
>
> The authors thank the reviewer for the careful review. Please see our response as follows:
>
> > This definition rather introduces restrictions on the class of MDPs considered.
>
> Agreed. In order for our method to be effective, the problem must contain the symmetry that is encoded by the equivariant model. SO2-symmetric problems are an important class of problems that are common in robotics.
>
> > My question is apart from visual tasks like those solved in this paper, can the authors provide examples and intuitions into when those assumptions hold? Is it the case that G-invariant MDPs are exclusive to image state representations?
>
> Our work focuses on visual state representations, but the G-invariant MDPs can be applied to other tasks. One intuitive example would be similar manipulation problems in a state space where the state contains the pose of the end-effector and the pose of all objects (e.g., the Fetch environments in OpenAI gym). Another example could be navigation problems where the state is a top-down image of the maze, and the action is the movements along the x-axis and y-axis. In general, this approach makes sense anytime the state space and the action space can be represented by features which transform under SE(2), that is, can be rotated and translated, and the assumptions in Definition 4.1 are satisfied.
>
> > What assumptions are we really making on the reward and transition functions? Do these relate to Lipshitzness at all?
>
> The only assumptions that we are making are expressed in Definition 4.1. -- that the transition function and the reward functions are invariant to the transformations by elements of the group. There are no assumptions about the smoothness of the reward function and the transition function.
>
> > Can we handle collision-avoidance scenarios for example?
>
> Though our experiments do not cover collision-avoidance scenarios, it should be possible to apply our method in such a setting because collisions would be invariant under the group operator, i.e. collision events should be invariant under rotations of the agent, the environment, and the obstacles.
>
> > Can we please see a demonstration in higher dimensional action spaces? Would the method apply to ATARI domains as well? Or is it that the type of discontinuities present in ATARI prohibit the application of the proposed method?
>
> Most Atari games don’t have the SO(2) symmetries considered here (with some exceptions, as discussed in Mondal et al. 2020), so our method would not be applicable there for that reason. The fact that ATARI games have discontinuities should not cause a problem.
>
> > It would be great to have a section discussing the limitations in terms of MPDs that could be considered when using the current approach.
>
> Thanks for pointing this out, we will discuss the limitations in the revision. The primary limitation is that our method requires the MDP to be a G-invariant MDP, i.e., to have an invariant reward function and an invariant transition function under the SO(2) group operator. In other words, our method can only be applied to problems that have SO(2) symmetries (although the method is probably also applicable to reflection, scaling, or other types of symmetries). Nevertheless, many robotics problems *do* have this symmetry because they are embedded in SE(3) state spaces.
>
> > I am a bit confused about the novelty of the current approach beyond applying equivariant architectures to reinforcement learning. Can the authors please help me understand the major contributions of this work?
>
> We believe our paper is the first work to apply equivariant models to standard visual control problems in robotics. We propose novel equivariant versions of SAC and LfD. While Mondal et al. (2020) have indeed proposed an equivariant version of DQN prior to our work, we are the first to show that equivariant models can significantly outperform baselines that use vanilla models.

---

### Official Review · Reviewer_nGwx · 2021-11-02

**Correctness:** 3
**Technical Novelty And Significance:** 3
**Empirical Novelty And Significance:** 3
**Recommendation:** 6
**Confidence:** 3

**Main Review:**

Strengths:

1. The proposed method and propositions are backed by solid theoretical proofs.

2. The claims made about the proposed method are validated through extensive experimentations and comparison against strong baselines.

3. The related works are well-covered. This paper is novel in its identification of equivariant properties of optimal Q-functions and optimal policies and its application to problems in robotics such as object manipulation (as opposed to previous applications to toy settings). It shows the improved performance of an equivariant DQN on these tasks and introduces an equivariant SAC model. It is closely related to Wang et al. (2021) which focuses on the translational and rotational invariance of Q-functions whereas this paper focuses on group equivariance.

4. Overall presentation of the paper is good

Weaknesses:

1. Is the claim of proposing an equivariant version of DQN as a novel contribution correct? The authors reference Mondal et al. (2020) which claims to introduce equivariant DQN. Is what the authors proposing different in nature from Mondal et al. (2020)? The two works seem to use different symmetry groups (this paper talks about SO(2) and Mondal et al. talks about E(2)) but this and any/all other differences need to be better highlighted in the paper.

2. The improvement in the reward shows the ability of the Equi DQN and Equi SAC to faster learn the policies but can it be ensured this is because of the inductive bias of the model architecture as claimed? One suggestion is to include a generalization experiment to test the robustness to equivariance where vanilla DQN fails. Additionally, the equivariant models are said to be applicable to rotationally symmetric problems. How would they perform in other problem settings without the symmetry assumption/ in the presence of other types of symmetry? Would they still beat the vanilla models?

3. In the experiment section, the baseline CNN DQN only says “DQN with conventional CNN instead of equivariant network”. Is there a reason why the Equi DQN and baseline DQN have different numbers of convolutional layers and is this a fair comparison? What is the significance of the extra two fully-connected layers? Does this mean the Equi DQN needs more parameters to learn compared to the baseline DQN. If so, then there should be a baseline DQN that has the same number of parameters.

4. The paper has minor grammatical errors/omitted words. A careful proof-reading should be helpful.



**Summary Of The Paper:**

This paper defines and theoretically characterizes a class of group-equivariant MDPs and studies its invariance and equivariance properties. It introduces two new models, one for discrete action spaces (equivariant DQN) and one for continuous action spaces (equivariant SAC). It compares the performance of the proposed methods against strong competitive baselines for multiple robotic manipulation tasks and establishes their superior performance.

**Summary Of The Review:**

The paper has an interesting premise about the advantage of encoding symmetry as an inductive bias in the data. It is novel in terms of introducing equivariant SAC but the claim of introducing equivariant DQN is ambiguous as it has already been introduced in previous work (Mondal et al. 2020).  Their experiments show improved performance of equi DQN and equi SAC over other methods for rotationally symmetric problems in robotic manipulation (which is different from previous works).  Although their results look very promising,  their claims may be better supported by additional experimentation  such as evaluating the performance of the proposed models in non-symmetric environments. Additionally, in order to ensure fair comparison, their equi DQN needs to be compared with a baseline DQN model with the same number of parameters.

---

> ### Author Response · Authors · 2021-11-17
> **Author Response to Reviewer nGwx**
>
> The authors thank the reviewer for the careful review. Please see our response as follows:
>
> > Is the claim of proposing an equivariant version of DQN as a novel contribution correct? The authors reference Mondal et al. (2020) which claims to introduce equivariant DQN. Is what the authors proposing different in nature from Mondal et al. (2020)? The two works seem to use different symmetry groups (this paper talks about SO(2) and Mondal et al. talks about E(2)) but this and any/all other differences need to be better highlighted in the paper.
>
> Our approach to DQN is fundamentally similar to that of Mondal et al. (2020). However, we apply the ideas in visual robotic applications, rather than just the two ATARI domains used by Mondal et al. (2020). Importantly, whereas Mondal et al. (2020) show only very modest improvements in sample efficiency, our results show that the equivariant model does significantly better on our problems. This is probably a significant finding in terms of evaluating the utility of the approach. We’ll clarify the above in the document.
>
> > The improvement in the reward shows the ability of the Equi DQN and Equi SAC to faster learn the policies but can it be ensured this is because of the inductive bias of the model architecture as claimed? One suggestion is to include a generalization experiment to test the robustness to equivariance where vanilla DQN fails.
>
> Thanks for this suggestion. We are currently working on a generalization experiment where we compare the ability for vanilla models and equivariant models to generalize over the symmetry group.
>
> > Additionally, the equivariant models are said to be applicable to rotationally symmetric problems. How would they perform in other problem settings without the symmetry assumption/ in the presence of other types of symmetry? Would they still beat the vanilla models?
>
> If the problem does not contain the symmetry encoded by the equivariant model, then our approach will be less effective. It is certainly possible to encode other types of symmetry, e.g., reflection symmetry, scaling symmetry, etc., but we have not explored this for RL problems like this. However, we believe it is fair to assume that other symmetries could be just as useful as the rotational symmetries considered here as long as they reflect symmetries that actually exist in the problem.
>
> > In the experiment section, the baseline CNN DQN only says “DQN with conventional CNN instead of equivariant network”. Is there a reason why the Equi DQN and baseline DQN have different numbers of convolutional layers and is this a fair comparison? What is the significance of the extra two fully-connected layers?
>
> The reason for having two extra convolutional layers is that we cannot use FC layers to flatten the feature map (as is in the baseline DQN) because the FC layers will break the equivariant property. Instead, we need to use those two convolutional layers to flatten the feature map in an equivariant way. In other words, those two convolution layers serve a similar purpose as the two FC layers in our baseline CNN’s architecture.
>
> > Does this mean the Equi DQN needs more parameters to learn compared to the baseline DQN. If so, then there should be a baseline DQN that has the same number of parameters.
>
> The baseline CNN has 10x the number of trainable parameters as the equivariant network (35M vs. 2.6M). If we look at the encoder parts of both networks (excluding the last two flattening layers of both networks), the baseline CNN’s encoder has 1.5x the number of trainable parameters as the equivariant network’s encoder (1.5M vs 1M). We view this as being unfairly biased in favor of the baseline. The equivariant model is much more parameter efficient.
>
> > The paper has minor grammatical errors/omitted words. A careful proof-reading should be helpful.
>
> Agreed. We’ll fix those problems.

---

### Official Review · Reviewer_27S2 · 2021-11-02

**Correctness:** 3
**Technical Novelty And Significance:** 2
**Empirical Novelty And Significance:** 3
**Recommendation:** 8
**Confidence:** 3

**Main Review:**

Strengths
+ Very clearly motivates sample efficiency + generalization and articulates inductive bias from the perspective of adding equivariance to the model vs doing data augmentation.
+ Results do show better data efficiency compared to baselines.

Weakness/Comments
- Results don't get into generalization but the claims/motivation in intro and related work leave the impression that this axis would be investigated,
- The exposition of the approach is a bit hard to parse and sprinkling in some intuition and ground things to the application would be helpful.
- What is the input? Is the kinds of images shown in Fig 5 or the depth map in Fig 2b? Some of the assumptions for this depth map don't make sense from a real application perspective: (i) if a depth camera is mounted near the robot end-effector it won't be possible to get the image in base frame, if it is overhead mounted then there will be occlusions so the object may not be always visible; (ii) depending on the task the gripper will often partially/fully occlude the object -- how much is this a problem? (iii) both Fig 5 and 2b are very idealized and in practice many other visual distractors would be present.
- The claim in Sec 7 about "transfers easily to real-world environments" is not well supported and given the points above (and possibly other) this is in fact going to be quite challenging. A more nuanced coverage of limitations would be helpful to constraint where this method works and where it doesn't, rather that surface level limitations of learning in simulation and delta state based action.

**Summary Of The Paper:**

This paper develops equivariant architecture for RL, specifically DQN and SAC. The theoretical exposition is general and a specific SO2 instantiation is shown to outperform baselines on image based RL tasks.


**Summary Of The Review:**

Bringing inductive bias to mostly unstructured policies is useful for RL. The results are promising, but some claims are overblown or not well supported.

---

> ### Author Response · Authors · 2021-11-17
> **Author Response to Reviewer 27S2**
>
> The authors thank the reviewer for the careful review. Please see our response as follows:
>
> > Results don't get into generalization but the claims/motivation in intro and related work leave the impression that this axis would be investigated
>
> Thanks for pointing this out. We plan to run a generalization experiment where the training environment only has one specific rotation while the testing environment has random rotations. We plan to add the result to the revision.
>
> > The exposition of the approach is a bit hard to parse and sprinkling in some intuition and ground things to the application would be helpful.
>
> We will try to make the description more intuitive in the revision.
>
> > What is the input? Is the kinds of images shown in Fig 5 or the depth map in Fig 2b?
>
> In our experiments, the input is the depth map like that shown in Figure 2b.
>
> > Some of the assumptions for this depth map don't make sense from a real application perspective: (i) if a depth camera is mounted near the robot end-effector it won't be possible to get the image in base frame, if it is overhead mounted then there will be occlusions so the object may not be always visible; (ii) depending on the task the gripper will often partially/fully occlude the object -- how much is this a problem? (iii) both Fig 5 and 2b are very idealized and in practice many other visual distractors would be present.
>
> We agree -- this is a little tricky to do in practice. If we just take a top-down depth image from overhead, then it would include the robot in the image which would break the equivariant assumptions. Instead, we do the following: 1) construct a point cloud from two fixed side-view cameras on the scene; 2) remove the robot from the point cloud; 3) project the point cloud onto a top down height map similar to what is shown in Fig 2b. Although it’s complicated, this method seems to work pretty well in practice. We’re currently implementing this on a physical robotic system, although that won’t be done in time for the end of the review period.
>
> > The claim in Sec 7 about "transfers easily to real-world environments" is not well supported and given the points above (and possibly other) this is in fact going to be quite challenging.
>
> We’re currently evaluating the method described above on a tabletop UR5 system in our lab and it seems to be feasible, although we agree that there are always challenges to any physical implementation.
>
> > A more nuanced coverage of limitations would be helpful to constraint where this method works and where it doesn't, rather that surface level limitations of learning in simulation and delta state based action.
>
> That’s a good point. The main limitation here is probably the constraint that our system satisfies the two G-equivariant conditions of Definition 4.1. It is precisely for this reason that we need to crop the robot from the image using the point cloud reprojection strategy described above. Nevertheless, we argue that the gains in sample efficiency are worth the extra effort required to reproject -- it’s not unusual to do some sort of image reprojection in many robotics applications anyway.

---

> > ### Comment · Reviewer_27S2 · 2021-11-29
> > **Thank you for the response**
> >
> > Thanks for the generalization experiment and acknowledging the limitations wrt implementation. While there is much more to do on that axis, I think given the revision this a promising start, so I am increasing my score from 6 to 8.

---

### Official Review · Reviewer_pbdx · 2021-11-03

**Correctness:** 4
**Technical Novelty And Significance:** 2
**Empirical Novelty And Significance:** 4
**Recommendation:** 8
**Confidence:** 4

**Main Review:**

I found the paper to be clearly written and generally well-executed. The main strength of the paper is the experimental results which are very supportive in a range of different tasks. Its main weakness is lack of novelty, on the theoretical side. This is because, as noted by the paper itself, equivariant networks have been used for learning equivariant policies and invariant Q functions in previous work. However, given the importance of this domain, I believe the focus on this group and proposed methodology still has interesting contributions (e.g., in learning from demonstrations).

Questions/comments:

Proposition 4.1 is not new, and proper references should give credit to early works on symmetric MDPs (e.g., “Symmetries and Model Minimization in Markov Decision Processes”)

On “preventing critic from becoming overconstrained”: while I understand the proposed argument, it doesn’t seem valid to me, since you have the option of having many equivariant layers before the scaled mean pooling operation. While using max pooling instead of min pooling is fine, the argument seems tangential. Any comments?

Since the results are surprisingly good compared to the model that replaces C8/C4 equivariant layer with ordinary convolution, could you please also provide the results for C2 equivariant layer? Such a layer would only have a factor of 2 improvement over the CNN (in data-efficiency) and therefore it would help make sense of the results.


**Summary Of The Paper:**

The paper uses rotation equivariant CNNs for model-free reinforcement learning. More specifically, it is argued that for robotic manipulation tasks the corresponding MDP is invariant under translation and rotation, and therefore one could more efficiently learn equivariant/invariant policy/value functions. This idea is applied to both DQN and SAC for control with finite and continuous action spaces respectively. Experimental results on several tasks demonstrate substantial improvement over the competing methods.


**Summary Of The Review:**

see above.

---

> ### Author Response · Authors · 2021-11-17
> **Author Response to Reviewer pbdx**
>
> The authors thank the reviewer for the careful review. Please see our response as follows:
>
> > Its main weakness is lack of novelty, on the theoretical side. This is because, as noted by the paper itself, equivariant networks have been used for learning equivariant policies and invariant Q functions in previous work.
>
> Agreed. However, most prior work is demonstrated in “toy” problems whereas here we apply to visual control problems in robotics. Also, our equivariant SAC and LfD are novel contributions. Equi-SAC is particularly interesting because, unlike DQN, that model does not require encoding action-values at the outputs of the network, thereby enabling us to use larger cardinality groups.
>
> > Proposition 4.1 is not new, and proper references should give credit to early works on symmetric MDPs (e.g., “Symmetries and Model Minimization in Markov Decision Processes”)
>
> Thanks for pointing this out. In the revision, we will add the reference and add a few sentences about how our “G-invariant MDPs” can be viewed as a special case of an MDP homomorphism, for which there exist long-standing proofs about the optimality of lifted policies. Our proof of the proposition in our paper is nice, however, because it’s short and easy to understand relative to the more general analysis for MDP homomorphism.
>
> > On “preventing critic from becoming overconstrained”: while I understand the proposed argument, it doesn’t seem valid to me, since you have the option of having many equivariant layers before the scaled mean pooling operation. While using max pooling instead of min pooling is fine, the argument seems tangential. Any comments?
>
> Thank you for pointing this out.  You are correct that a network $f$ with several equivariant layers before a scaled mean pooling operation would also break the undesired symmetry $f(gx,y) = f(x,y)$ while preserving the desired symmetry $f(gx,gy) = f(x,y)$.  However, using a linear layer (scaled mean pooling) as opposed to a non-linear layer to perform the projection from regular representation type to trivial representation type would still enforce additional symmetry on the latent representation.
>
> Specifically, assuming the input and each hidden layer are several copies of the regular representation there would not be any overconstraint in the mapping from input to the final hidden layer.  However, since the final output is of trivial representation type, it would then be necessary to map from several copies of the regular representation to the trivial representation $h : \rho_{reg}^k \to \rho_{triv}$.  If this were done with a linear scaled mean pooling operation, the overconstraint would then apply at this stage.  That is $h(g v_1, v_2, \ldots,v_k) =  h( v_1, v_2, \ldots,v_k)$.   A max pooling layer, however, avoids imposing additional symmetry constraints.
>
> We will edit the section to make the potential issue more clear.
>
> > could you please also provide the results for C2 equivariant layer? Such a layer would only have a factor of 2 improvement over the CNN (in data-efficiency) and therefore it would help make sense of the results.
>
> Thanks for pointing this out. We are trying to run this experiment and add this baseline in the revision.

---

### Official Review · Reviewer_cinR · 2021-11-08

**Correctness:** 3
**Technical Novelty And Significance:** 3
**Empirical Novelty And Significance:** 3
**Recommendation:** 8
**Confidence:** 3

**Main Review:**

Strength:

- Sound theory of group-invariant MDP. The theory also characterizes both the invariance and equivariance cases.
- Applications on realistic robotic manipulation
- Showcasing on different RL (SAC and DQN) and robot learning (LfD) algorithms.

Weakness:
- The description sometimes is hard to follow


In general, the paper pursues an interesting research problem. it's well written. The proposed idea address the main well-understood challenge in robot learning. I would have only following comments.

Regarding to the invariant-MDP, the proposed approach only deals with visual state spaces, so is it possible to extend to include proprioceptive information in the state space?

Is the group element $G$ pre-defined and does it contain only a single rotation operator $g$ as defined in Experiment? If so would it be too simplistic? More ablation studies regarding this choice would be more helpful.


**Summary Of The Paper:**

This paper explores to integrate equivariance deep learning to robotic applications. The authors propose two main contributions: 1) define and theoretically characterize an important class of group-equivariant MDPs, 2) and integrate equivariant variations to DQN, SAC, and LfD. The paper provides many different sets of experiments. The results are promising which shows benefits of the proposed approach.


**Summary Of The Review:**

The paper proposes interesting ideas that might be useful for robot learning tasks. Experiment results are positive. There are some questions as raised in the above section.

---

> ### Author Response · Authors · 2021-11-17
> **Author Response to Reviewer cinR**
>
> The authors thank the reviewer for the careful review. Please see our response as follows:
>
> > The description sometimes is hard to follow
>
> Thanks for pointing that out. We will do our best to make our explanations of the model more intuitive in the revision.
>
> > Regarding to the invariant-MDP, the proposed approach only deals with visual state spaces, so is it possible to extend to include proprioceptive information in the state space?
>
> We’re assuming “proprioceptive” refers to the joint space of the robot. It is unclear whether our approach is directly applicable in joint space because it is unclear what the SO(2) symmetries would be in the joint space. Nevertheless, there are likely some kinds of symmetries there, but we haven’t explored this yet.
>
> > Is the group element $G$ pre-defined and does it contain only a single rotation operator $g$ as defined in Experiment? If so would it be too simplistic? More ablation studies regarding this choice would be more helpful.
>
> This paper focuses on a discretized version of the symmetry group SO(2), which means that our model is simultaneously equivariant to every rotation operator in the group (either multiples of $\pi/2$ or $\pi/4$).  We focus on rotationally symmetric problems, but the concept can be applied to other types of symmetry, e.g., reflection. We believe that rotational symmetry is sufficiently important, especially in the field of robotic manipulation, to warrant our focus.

---

### Author Response · Authors · 2021-11-20
**Summary of Revision**

The authors thank all reviewers for the careful review. We have made several changes in the revision based on the reviewer’s suggestions to make our paper stronger. We have uploaded the revision with the major changes labeled in blue. Here is a summary of the changes:
- Added a generalization experiment in Section 6.5 to evaluate the generalizability of the equivariant network.
- Added the reference to MDP homomorphism and a discussion about its relationship with Proposition 4.1.
- Revised the baseline CNN architectures to have the approximately same amount of parameters as our equivariant architectures.
- Revised our claim about “proposing Equivariant DQN”, highlighted the difference of our version of Equivariant DQN compared with Mondal et.al (2020) in Section 5.1
- Revised Section 7 to talk about the limitation in terms of the restriction of $G$-invariant MDPs.
- Added an ablation study of the equivariant network defined in group $C_2$ in Appendix H.2.
- Added an ablation study of using the equivariant network in non-symmetric tasks in Appendix H.3.
- Clarified the potential confusion in “Preventing the critic from becoming overconstrained”.
- Fixed the grammar issues.

---

### Decision · Program_Chairs · 2022-01-20

**Decision:**

Accept (Spotlight)

**Comment:**

The paper investigates the use of equivariant neural network architectures for model-free reinforcement learning in the context of visuomotor robot manipulation tasks, exploiting rotational symmetries in an effort to improve sample efficiency. The paper first provides a formal definition and theoretical evaluation of a class of MDPs for which the reward and transition are invariant to group elements ("group-invariant MDPs"). It goes on to describe equivariant versions DQN, SAC, and learning from demonstration (LfD). Experiments on a set of different manipulation tasks reveal that the proposed architectures outperform contemporary baselines in terms of sample complexity and generalizability, while ablations demonstrate the contribution of the different model components.

The idea of structuring a neural architecture to exploit symmetry present in a domain as a means of improving sample complexity is compelling and principled. The contributions of the paper are two-fold. First, while the idea of exploiting symmetry in the context of deep RL is not new, the paper describes a variation of equivariant DQN that is effective for visual control domains (visuomotor control) that are more challenging and realistic than those considered previously. Second, the paper proposes novel equivariant versions of SAC and LfD and validates their effectiveness through extensive experiments. Following a detailed author response to the initial reviews together with the inclusion of additional experiments and other updates to the paper, the reviewers largely agree on the significance of these contributions and value of the paper as a whole.